# Gender Differences in Patients with Metastatic Pancreatic Cancer Who Received FOLFIRINOX

**DOI:** 10.3390/jpm11020083

**Published:** 2021-01-30

**Authors:** Jinkook Kim, Eunjeong Ji, Kwangrok Jung, In Ho Jung, Jaewoo Park, Jong-Chan Lee, Jin Won Kim, Jin-Hyeok Hwang, Jaihwan Kim

**Affiliations:** 1Department of Internal Medicine, Seoul National University Bundang Hospital, Seoul National University College of Medicine, Seongnam 13620, Korea; newscboy@naver.com (J.K.); herojkr@hanmail.net (K.J.); andpassion@naver.com (I.H.J.); jaewoo0604@naver.com (J.P.); ljc0316@naver.com (J.-C.L.); jwkim@snubh.org (J.W.K.); jhhwang@snubh.org (J.-H.H.); 2Medical Research Collaborating Center, Seoul National University Bundang Hospital, Seoul National University College of Medicine, Seongnam 13620, Korea; 99145@snubh.org

**Keywords:** pancreatic cancer, FOLFIRINOX, chemotherapy, gender

## Abstract

Background: The combination of 5-fluorouracil, leucovorin, irinotecan, and oxaliplatin (FOLFIRINOX) is a very effective chemotherapeutic regimen for unresectable pancreatic cancer. Previous studies have reported that female gender may be a predictor of a better response to FOLFIRINOX. This study was aimed at investigating the clinical outcomes and dose modification patterns of FOLFIRINOX by gender. Methods: Patients with metastatic pancreatic cancer (MPC) who began FOLFIRINOX as the first-line therapy at Seoul National University Bundang Hospital between 2013 and 2018 were enrolled. The patients received at least four chemotherapy cycles. Local regression and a linear mixed model were used to analyze dose modification patterns by gender. Results: Ninety-seven patients with MPC (54 men; 43 women) were enrolled. In the first FOLFIRINOX cycle, there were significant differences in age and body surface area between the genders (58.8 (men) and 64.9 years (women), *p* = 0.005; 1.7 (men) and 1.6 m^2^ (women), *p* < 0.001, respectively). The median progression-free survival (PFS) and overall survival (OS) were 10.8 and 18.0 months, respectively. There was a trend of longer PFS (10.3 (men) and 11.9 months (women), *p* = 0.153) and a significantly longer OS (17.9 (men) and 25.9 months (women), *p* = 0.019) in female patients. During the first year of FOLFIRINOX treatment, there was a significant difference of the age-corrected dose reduction pattern by gender (a mean of 95.6% dose at the initial cycle and −0.35% of dose reduction per week in men versus a mean of 90.7% dose at the initial cycle and −0.53% of dose reduction per week in women, *p*-value of the slope: <0.001). There was no difference in the adverse event rates between the genders. Conclusions: Female patients showed longer OS despite a more rapid dose reduction during each cycle. Gender differences should be considered during FOLFIRINOX treatment.

## 1. Introduction

Pancreatic cancer is a lethal malignancy and the fourth leading cause of cancer-related death in the United States, with a current 5-year survival rate of only approximately 9% [1]. In up to 85% of patients, pancreatic cancer is diagnosed at an advanced stage because of infiltration of the surrounding vessels or distant metastasis [2]. In patients with metastatic pancreatic cancer (MPC), the combination of 5-fluorouracil (5-FU), leucovorin, irinotecan, and oxaliplatin (FOLFIRINOX) or gemcitabine and nab-paclitaxel resulted in significantly longer overall survival (OS) than that associated with gemcitabine monotherapy [3,4]. Furthermore, FOLFIRINOX has shown excellent efficacy in not only palliative but also adjuvant settings [5], and this regimen has been used as the neoadjuvant chemotherapy in several phase III clinical trials [6]. The mechanism of FOLFIRINOX is regarded as synergistic activity of irinotecan when it is administered before fluorouracil and leucovorin and synergistic activity of irinotecan and oxaliplatin [3]. However, the regimen has been associated with a high incidence of adverse events, including grade 3 or 4 neutropenia and fatigue.

The effects of gender in cancer treatment are not generally considered in preclinical experiments, clinical trials, or real clinical settings. However, there are differences in the efficacies and toxicities of chemotherapeutic agents between male and female patients [7]. For example, 5-FU, which is the backbone of the FOLFIRINOX regimen, degraded more slowly and was associated with higher toxicity in female patients [8,9]. In addition, a previous study reported that female gender may be a predictor of a better response to FOLFIRINOX. In this study, female gender was associated with a significantly higher disease control rate and showed a tendency towards a longer median progression-free survival (PFS) [10]. However, a secondary analysis within the PRODIGE 4/ACCORD 11 trial did not conclusively show a possible effect of gender on the prognosis of patients receiving FOLFIRINOX, because median OS and PFS results in female were superior to those in males, without statistical significance [11]. Therefore, the association remains controversial and elucidation is necessary for further evaluation of the effects of gender. The current study was aimed at investigating gender differences in clinical outcomes and dose modification patterns during FOLFIRINOX chemotherapy in patients with MPC.

## 2. Materials and Methods

### 2.1. Patients

Patients with MPC who received the first-line FOLFIRINOX treatment between January 2013 and December 2018 at the Seoul National University Bundang Hospital were retrospectively included. Metastatic pancreatic cancer is defined as cancer that started in the pancreas and spread to areas outside the pancreas, such as the liver, peritoneum, lungs, or distant lymph nodes. The exclusion criteria were as follows: (1) less than four cycles of FOLFIRINOX due to treatment intolerability, adverse events, or a loss to follow up; (2) resectable, borderline resectable, or locally advanced pancreatic cancer at the time of diagnosis; (3) use of FOLFIRINOX as the second-line or later chemotherapy; (4) a history of radiation therapy prior to FOLFIRINOX use; and (5) an Eastern Cooperative Oncology Group Performance Score (ECOG PS) of 2 or higher. The clinical and pathological records of the patients were obtained from a retrospective review of electronic medical records and pathologic reports. Treatment response was evaluated according to response evaluation criteria in solid tumors (RECIST) and the carbohydrate antigen 19 (CA 19-9) was used as a supplementary tool. This study was approved by the institutional review board of Seoul National University Bundang Hospital (IRB# B-1907/550-112).

### 2.2. Calculation of the Modified Dose of FOLFIRINOX

The FOLFIRINOX regimen was administered in 14 day cycles according to the PRODIGE 4/ACCORD 11 trial [3], with dose reduction or increases of the intervals between cycles decided by a physician. The response to chemotherapy was evaluated every 8 to 12 weeks by using contrast-enhanced computerized tomography and by determining the carbohydrate antigen 19 (CA 19-9) levels. Magnetic resonance imaging or positron emission tomography was also used for evaluation if necessary. FOLFIRINOX administration was continued until the patients showed disease progression or treatment intolerability. The relative dose intensity (RDI) of FOLFIRINOX was defined according to its definition in a previous study performed by our group (Figure 1) [12], in which a modified Hryniuk calculation method was used [13]. Single-agent RDI is the simple proportion of the actual dose delivered compared to the standard dose of each agent (85, 180, 400, and 2400 mg/m^2^ for oxaliplatin, irinotecan, 5-FU as a bolus, and 5-FU via continuous intravenous injection, respectively), and the multi-drug RDI was the mathematical average of single-agent RDIs. 

### 2.3. Study Objectives

The primary outcomes were OS and progression-free survival (PFS). The secondary outcomes were the dose modification pattern of FOLFIRINOX according to time and adverse events. Data on adverse events were collected according to the National Cancer Institute Common Terminology Criteria for Adverse Events version 5.0.

### 2.4. Statistical analysis

All statistical analyses were performed using R version 3.6.2. Chi-square or Fisher’s exact test was used for a comparative analysis of categorical data. According to the time, the dose modification pattern of FOLFIRINOX was evaluated with a local regression method (locally estimated scatterplot smoothing—LOESS), namely a linear mixed model. A LOESS plot is a non-parametric regression method that combines multiple regression models. To identify the marginal effect of time, the linear mixed model was adjusted by the average effect of age. Univariable analyses for OS and PFS were performed using the Kaplan–Meier method with log-rank tests. Statistical significance was defined as *p* < 0.05.

## 3. Results

### 3.1. Baseline Patient and Tumor Characteristics

A total of 97 patients with MPC (54 men and 43 women) were enrolled in this study. The baseline characteristics of the patients are summarized in Table 1. The median age at diagnosis was 61.1 years (range, 41.0–85.5 years). The male patients were significantly younger than the female patients (58.8 and 64.9 years, respectively; *p* = 0.005). Furthermore, the male patients had a significantly larger body surface area than did the female patients (1.7 and 1.6 m^2^, respectively; *p* < 0.001). However, there were no differences in other characteristics such as the location of the primary tumor, metastatic sites, body mass index, initial CA 19-9 level, or ECOG PS between the genders. Granulocyte colony-stimulating factor was used more than once in 84 patients (86.6%). During chemotherapy, five patients received surgery as following: two patients received pylorus-preserving pancreaticoduodenectomy, two received distal pancreatectomy, and one received explorative laparotomy. After progression was noted, despite FOLFIRINOX, 50 patients (51.5%) received the second-line chemotherapy: 40 patients (41.2%) received gemcitabine-based combination chemotherapy and 10 (10.3%) received TS-1.

### 3.2. PFS and OS

The median PFS and OS in this study were 10.8 and 18.0 months, respectively. The median PFS values for the male and female patients were 10.3 and 11.9 months, respectively; the intergroup difference was not significant (*p* = 0.153, Figure 2A). The median OS values for the male and female patients were 17.9 and 25.9 months, respectively; the intergroup difference was not significant (*p* = 0.019, Figure 2B). The 1- and 2-year survival rates in male patients were 61.3% and 23.6%, respectively; the corresponding percentages in female patients were 72.9% and 56.8%. Four patients (one male and three female patients) survived for more than 3 years. Of these, three patients showed local invasion of other organs (e.g., the stomach or spleen) and peritoneal seeding at the time of diagnosis and one patient showed a few tiny hepatic metastases.

### 3.3. FOLFIRINOX Dose Modification Pattern

During a year of FOLFIRINOX chemotherapy, the dose modification pattern was determined with regression analyses. According to the LOESS regression, there was a difference in the dose reduction patterns of chemotherapy in a year between male and female patients (Figure 3A). For statistical comparison of the dose reduction patterns between male and female patients, the linear mixed model was used. Before correction for age, male patients received a mean FOLFIRINOX dose of 95.8% as the initial cycle and the slope of dose reduction was −0.33% per week (*p*-value of the slope: <0.001; Figure 3B). However, female patients received a mean dose of 89.8% and the slope was −0.52% per week. After correction for mean age (61.2 years old), the slopes for male and female patients were −0.35% and −0.53% per week, respectively (*p*-value of the slope: <0.001); and the doses for the male and female patients in the initial cycle were 95.6% and 90.7%, respectively (*p*-value of the slope: <0.001; Figure 3C).

### 3.4. Treatment-Related Adverse Events and the Number of Visits to the Emergency Department

During a year of chemotherapy, the most common grade 3 or 4 chemotherapy-related adverse event was neutropenia (Table 2). Nausea, febrile neutropenia, and sensory neuropathy also occurred in more than 10% of the patients. There was no difference in grade 3 or 4 treatment-related adverse events between male and female patients. Due to the chemotherapy-related adverse events, 36 patients had to visit the emergency department and five patients visited the emergency department more than three times (Table 3). However, there was no difference in the number of visits to the emergency department between male and female patients.

## 4. Discussion

FOLFIRINOX is a 5-FU–based combination chemotherapeutic regimen and is very effective for patients with unresectable pancreatic cancer [3]. However, the regimen is associated with considerable grade 3 or 4 toxicities, and dose modification during the treatment is very common. The current study aimed to assess differences in FOLFIRINOX outcomes by gender with regard to not only efficacy but also the amount of chemotherapeutic agents delivered.

Factors such as tumor biology, the immune system, body composition, and drug disposition differ between genders. These differences are associated with sex chromosomes, the levels of sex hormones, and environmental factors such as nutrition and microbiota [14]. The incidences of several cancers differ by gender, including esophageal and colorectal cancers. Besides differences in incidence and tumor location, drug pharmacology also differs. Fat-free body mass is approximately 80% and 65% in male and female patients, respectively; body composition also differs between the genders [15]. However, the current doses of chemotherapy are based on body surface area or body mass index, and gender is not considered in the calculation of chemotherapy doses [14].

5-FU is a drug with substantial inter-individual variability in clearance; the impact of gender on 5-FU clearance is significant, with the exposure in female patients being 26% higher than that in male patients [16]. A previous study showed that women were at a higher risk of grade 3 or 4 hematologic toxicities of 5-FU, and the higher clearance of 5-FU in men likely explains the higher toxicity of 5-FU in women with colorectal cancer [9]. It was also reported that the clearance of irinotecan in female patients is 30 to 38% less than that in male patients [17,18,19]. Consequently, adverse events were more frequently observed in women for most regimens [20].

A recent study about the FOLFIRINOX regimen by gender within the PRODIGE 4/ACCORD 11 trial reported longer median OS (13.1 vs. 10.3 months) and PFS (7.2 vs. 5.9 months) in women, although the differences between the genders were not significant (*p* = 0.101 and 0.169, respectively) [11]. Similar outcomes were also reported in another study, which showed a longer tendency of PFS (5.0 vs. 3.0 months, *p* = 0.099) and a higher disease control rate (91.7 vs. 48.0%, *p* = 0.001) in women [10]. Although these studies failed to conclusively show a definite effect of gender on the FOLFIRINOX regimen, they suggested the possibility of an effect. Compared to these studies, in this study, the median OS in the female patients was significantly longer than that in the male patients (25.9 months vs. 17.9 months, *p* = 0.019), despite the gender-related differences in PFS being non-significant (11.9 in female patients vs. 10.3 months in male patients, *p* = 0.153). Therefore, our results also suggested the possibility of better outcomes in female patients who received the FOLFIRINOX regimen. Additional research is necessary to determine whether these different outcomes of the FOLFIRINOX regimen are attributable to gender.

In addition to the chemotherapy response, this study focused on dose modification. Compared to previous studies, which only presented the average or median dose of FOLFIRINOX [12], in this study we calculated the modified dose for each cycle and compared the pattern of dose modification by gender according to time. As a result, female patients on average received 90% of the original dose in the first cycle and 65% of the original dose at 1 year. The doses seemed vastly different from those administered to male patients, who on average received 95% of the original dose in the first cycle and 83% of the original dose at 1 year. For statistical comparison of the dose modification pattern, a linear mixed model was adopted and the negative slopes were significantly different by gender. Considering that there were no differences in the rates of grade 3 or 4 adverse event or number of visits to the emergency department due to chemotherapy-related adverse events, our study suggested that better outcomes could be expected in female patients, even with smaller doses of FOLFIRINOX.

There were several limitations of this study. First, it was a retrospective single-center study. Therefore, further studies are necessary for the generalization of our results. Second, the median OS and PFS in this study seemed longer than those in previous studies because the current study excluded patients who underwent fewer than four cycles of FOLFIRINOX for the comparison of the dose modification pattern. Therefore, there was a possibility of selection bias for a good response in both genders. Third, no pharmacodynamic or pharmacokinetic data were available for the regimen. Lastly, half of the patients received second-line chemotherapy, which was mainly a gemcitabine-based regimen. Therefore, there is a possibility that second-line therapy might contribute to a difference in OS, although the regimens were heterogeneous. In conclusion, female patients showed better survival outcomes in spite of greater reductions in the doses of FOLFIRINOX in this study, and more attention should be focused on the effect of gender on FOLFIRINOX treatment in patients with MPC.

## Figures and Tables

**Figure 1 jpm-11-00083-f001:**
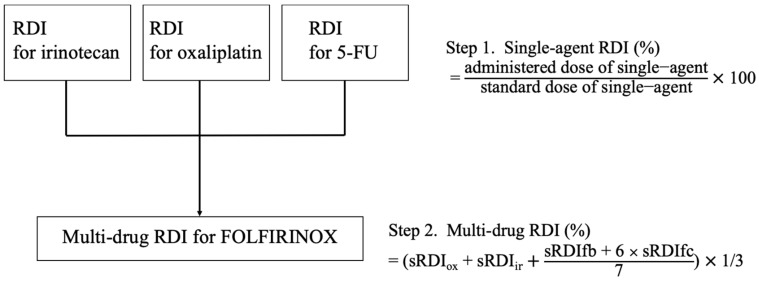
The modified Hryniuk model. 5-FU could be divided into sRDIfb and sRDIfc. The effect of the administration was not considered, which means that only the dose (mg) was used for calculations. RDI, relative dose intensity; ox, oxaliplatin; ir, irinotecan; fb, 5-FU bolus; fc, 5-FU continuous intravenous (Figure modified from reference 12).

**Figure 2 jpm-11-00083-f002:**
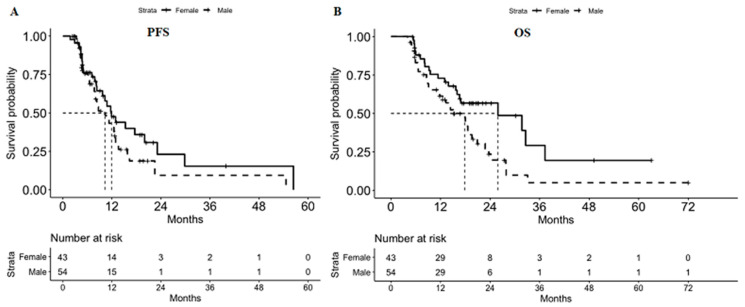
Median progression-free survival (**A**) and overall survival (**B**). (**A**) The median PFS values for male and female patients were 10.3 and 11.9 months, respectively (*p* = 0.153). (**B**) The median OS values for male and female patients were 17.9 and 25.9 months, respectively (*p* = 0.019). PFS, progression-free survival; OS, overall survival.

**Figure 3 jpm-11-00083-f003:**
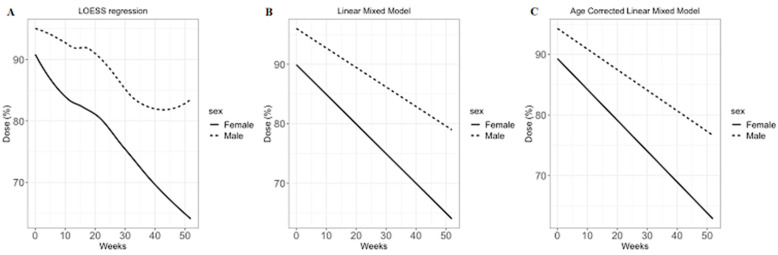
FOLFIRINOX dose modification model. There was a significant difference in the slopes between plots B and C (*p* < 0.001). (**A**) Local regression (LOESS) plot. (**B**) Linear mixed model without age correction. (**C**) Linear mixed model with marginal effect of time (corrected by mean age).

**Table 1 jpm-11-00083-t001:** Baseline characteristics of the patients.

	Men (*N* = 54)	Women (*N* = 43)	Total (*N* = 97)	*p*
Age, years				0.005
Median	58.8	64.9	61.1	
Range	55.0–64.2	58.7–69.9	55.9–68.7	
Tumor location, no. (%)				0.211
Head	19 (35.8)	21 (48.8)	40 (41.7)	
Body	7 (13.2)	6 (14)	13 (13.5)	
Tail	26 (49.1)	13 (30.2)	39 (40.6)	
Multiple	1 (1.9)	3 (7)	4 (4.2)	
Metastatic site, no. (%)				0.704
Liver	30 (37.0)	23 (36.5)	53 (36.8)	
Peritoneum	19 (23.5)	10 (15.9)	29 (20.1)	
Lung	7 (8.6)	9 (14.3)	16 (11.1)	
Lymph node	22 (27.2)	19 (30.2)	41 (28.5)	
Bone	3 (3.7)	2 (3.2)	5 (3.5)	
BMI (kg/m^2^)				0.248
Median	22.7	24.1	23.1	
Range	20.5–24.9	20.8–26	20.7–25.7	
BSA (m^2^)				<0.001
Median	1.7	1.6	1.6	
Range	1.6–1.8	1.5–1.6	1.5–1.8	
CA 19-9 (U/mL)				0.413
Median	900.0	620.0	760.0	
Range	192.6–3800.0	64.0–2100.0	118.0–2100.0	
ECOG PS score (%)				0.326
0	21 (38.9)	21 (48.8)	42 (43.3)	
1	33 (61.1)	22 (51.2)	55 (56.7)	
Use of G-CSF				0.647
Yes	46 (85.2)	38 (88.4)	84 (86.6)	
No	8 (14.8)	5 (11.6)	13 (13.4)	
Surgery (%)				0.793
Yes	2 (3.7)	3 (7.0)	5 (5.2)	
No	52 (96.3)	40 (93.0)	92 (94.8)	
Second-line chemotherapy, no. (%)				0.733 *
Gemcitabine	4 (7.4)	3 (7.0)	7 (7.2)	
Gemcitabine plus erlotinib	8 (14.8)	4 (9.3)	12 (12.4)	
Gemcitabine plus cisplatin	4 (7.4)	2 (4.7)	6 (6.2)	
Gemcitabine plus nab-paclitaxel	8 (14.8)	7 (16.3)	15 (15.5)	
TS-1	3 (5.6)	7 (16.3)	10 (10.3)	
None	27 (50.0)	20 (46.5)	47 (48.5)	

Table 1. BMI, body mass index; BSA, body surface area; CA 19-9, carbohydrate antigen 19-9; ECOG PS, Eastern Cooperative Oncology Group performance score; G-CSF, granulocyte colony-stimulating factor; *, second-line chemotherapy vs. no additional chemotherapy.

**Table 2 jpm-11-00083-t002:** Treatment-related grade 3 or 4 adverse events.

	Men (*N* = 54)	Women (*N* = 43)	Total (*N* = 97)	*p*
Hematologic				
Neutropenia	17 (31.5)	20 (46.5)	37 (38.1)	0.192
Febrile neutropenia	6 (11.1)	9 (20.9)	15 (15.5)	0.296
Anemia	0 (0.0)	1 (2.3)	1 (1.0)	0.909
Thrombocytopenia	1 (1.9)	4 (9.3)	5 (5.2)	0.235
Non-hematologic				
Anorexia	1 (1.9)	1 (2.3)	2 (2.1)	>0.99
Nausea	12 (22.2)	8 (18.6)	20 (20.6)	0.853
Vomiting	6 (11.1)	3 (7.0)	9 (9.3)	0.730
Diarrhea	4 (7.4)	2 (4.7)	6 (6.2)	0.892
Fatigue	0 (0.0)	2 (4.7)	2 (2.1)	0.378
Sensory neuropathy	7 (13.0)	3 (7.0)	10 (10.3)	0.531

**Table 3 jpm-11-00083-t003:** Number of visits to the emergency department due to chemotherapy-related adverse events.

	Men (*N* = 54)	Women (*N* = 43)	Total (*N* = 97)	*p*
				0.239
0	36 (66.7)	25 (58.1)	61 (62.9)	
1~2	17 (31.5)	14 (32.6)	31 (32.0)	
More than 3	1 (1.9)	4 (9.3)	5 (5.2)

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
