# Peer review of "Gender Differences in Patients with Metastatic Pancreatic Cancer Who Received FOLFIRINOX"

_jpm, 2021, doi:10.3390/jpm11020083_

Round 1
Reviewer 1 Report
Well conducted study despite the low number of enrolled patients
a few observations:
- In my opinion the definition of "metastatic pancreatic cancer" is approximate and should be better specified: liver, peritoneal or lung metastases
- the trend of tumor markers should be clearly written
- a part from PFS, quality of life, compliance to therapy must be considered
Author Response
Response to Reviewer 1 Comments
Well conducted study despite the low number of enrolled patients. a few observations:
Point 1: In my opinion the definition of "metastatic pancreatic cancer" is approximate and should be better specified: liver, peritoneal or lung metastases
Response 1: Thank you for your important comment. Metastatic pancreatic cancer is defined as the cancer that started in the pancreas and spread to outside pancreas such as other organ or peritoneum or distant lymph nodes. In this paper, we showed the most common metastatic sites in Table 1. We added the definition in Patients section in ‘Materials and methods’ (page 4 of 7).
Before:
Patients with MPC who received the first-line FOLFIRINOX between January 2013 and December 2018 at the Seoul National University Bundang Hospital were retrospectively included.
After:
Patients with MPC who received the first-line FOLFIRINOX between January 2013 and December 2018 at the Seoul National University Bundang Hospital were retrospectively included. Metastatic pancreatic cancer is defined as the cancer that started in the pancreas and spread to outside pancreas such as liver, peritoneum, lung, or distant lymph nodes.
Point 2: the trend of tumor markers should be clearly written
Response 2: Thank you for your critical comments. As you know very well, CA 19-9 is the most established tumor marker in pancreatic cancer and used for the treatment response. In this study, we also used CA 19-9 as the supplementary indicator for evaluation of the treatment response and presented initial CA 19-9 at diagnosis in Table 1. However, it was not elevated in some patients such as patients with Lewis Ag (-) or elevated in some conditions such as jaundice or cholangitis. Therefore, it is very difficult to show the trend and such a difficulty was common in other studies. Instead of the showing the trend, we added the comment about the role of CA 19-9 in Patients section in ‘Materials and methods’ (page 4 of 7)
Before:
The clinical and pathological records of the patients were obtained from a retrospective review of electronic medical records and pathologic reports.
After:
The clinical and pathological records of the patients were obtained from a retrospective review of electronic medical records and pathologic reports. Treatment response was evaluated according to response evaluation criteria in solid tumors (RECIST) criteria and the carbohydrate antigen 19 (CA 19-9) was used as a supplementary tool.
Point 3: a part from PFS, quality of life, compliance to therapy must be considered
Response 3: We appreciate your thoughtful comment. We totally agreed with your comment. Besides the biological differences, there are so many differences such as thinking about the quality of life or compliance to chemotherapy between men and women. However, we approached the differences between men and women in an aspect of objective measurement and they are beyond the focus of this study. Therefore, we did not change anything in this paper. Because we have interest in the sex difference, they could be a topic in our next study.

Reviewer 2 Report
In this paper by Kim et al, the authors explore if gender of the patient living with unresectable pancreatic cancer could serve a predictor of clinical outcomes and dose modification patterns when using the dosing regimen of FOLFIRINOX.
Overall, the publication does a good job of describing the background, rationale, analysis strategy and observed results. However, there are several suggestions with regards to their manuscript, as described below:
- The authors can increase the scope of their analysis by assessing the joint effect of gender with other clinical variables such as tumor stage, location or biomarkers like CA 19-9 using a multivariate logistic regression model.
- Instead of using “sex”, I suggest using “gender” as an alternative nomenclature to have complete clarity of the characteristic being discussed here (male/female).
- Their Introduction section can be more elaborate on certain points; for instance: 1. mechanism of action of chemotherapeutic drugs in FOLFIRINOX cocktail. 2. Provide 1-2 more lines with study-specific details on the two studies they cite (ref. 10 and 11).
Author Response
In this paper by Kim et al, the authors explore if gender of the patient living with unresectable pancreatic cancer could serve a predictor of clinical outcomes and dose modification patterns when using the dosing regimen of FOLFIRINOX.
Overall, the publication does a good job of describing the background, rationale, analysis strategy and observed results. However, there are several suggestions with regards to their manuscript, as described below:
Point 1: The authors can increase the scope of their analysis by assessing the joint effect of gender with other clinical variables such as tumor stage, location or biomarkers like CA 19-9 using a multivariate logistic regression model.
Response 1: Thank you for your important comment. We totally agree with your comment about assessing the joint effect of gender with other clinical variable using a multivariate logistic regression model. However, we want to show the different decreasing pattern of FOLFIRINOX instead of the total cumulative dose and it should be considered that there was a consideration of time during the treatment. The pattern of decreasing FOLFIRINOX dose according to time could not be incorporated into logistic regression model with other variables. In addition, you can find no difference of tumor stage, location, or CA 19-9 in table 1. Only age and BSA were different between male and female and BSA was already reflected in chemotherapy dose. Therefore, I would like to ask you to understand the aim of our study. We did not change anything in this paper.
Point 2: Instead of using “sex”, I suggest using “gender” as an alternative nomenclature to have complete clarity of the characteristic being discussed here (male/female).
Response 2: Thank you for your critical comments. We agreed with your opinion and exchangeed “sex” for “gender”.
Point 3: Their Introduction section can be more elaborate on certain points; for instance: 1. mechanism of action of chemotherapeutic drugs in FOLFIRINOX cocktail. 2. Provide 1-2 more lines with study-specific details on the two studies they cite (ref. 10 and 11).
Response 3: Thank you for your great suggestion. According to your suggestion, we added the mechanism of FOLFIRINOX regimen and details of the ref. 10 and 11 studies in Introduction.
Before:
Pancreatic cancer is a lethal malignancy and the fourth leading cause of cancer-related death in the United States, with a current 5-year survival rate of only approximately 9%.[1] In up to 85% of patients, pancreatic cancer is diagnosed at an advanced stage because of infiltration of the surrounding vessels or distant metastasis.[2] In patients with metastatic pancreatic cancer (MPC), the combination of 5-fluorouracil (5-FU), leucovorin, irinotecan, and oxaliplatin (FOLFIRINOX) or gemcitabine and nab-paclitaxel resulted in significantly longer overall survival (OS) than that associated with gemcitabine monotherapy.[3,4] Furthermore, FOLFIRINOX has shown excellent efficacy in not only palliative but also adjuvant settings,[5] and this regimen has been used as the neoadjuvant chemotherapy in several phase III clinical trials.[6] However, the regimen has been associated with a high incidence of adverse events including grade 3 or 4 neutropenia and fatigue.
The effects of gender in cancer treatment are not generally considered in preclinical experiments, clinical trials, or real clinical settings. However, there are differences in the efficacies and toxicities of chemotherapeutic agents between male and female patients.[7] For example, 5-FU, which is the backbone of the FOLFIRINOX regimen, degraded more slowly and was associated with higher toxicity in female patients.[8,9] In addition, a previous study reported that female gender may be a predictor of a better response to FOLFIRINOX. [10] However, a secondary analysis within the trial PRODIGE 4/ACCORD 11 did not conclusively show a possible effect of gender on the prognosis of patients receiving FOLFIRINOX because median OS and PFS in female was superior than those in male without statistical significance.[11] Therefore, the association remains controversial and elucidation is necessary for further evaluation of the effect of gender. The current study was aimed at investigating gender differences in clinical outcomes and dose modification patterns during FOLFIRINOX chemotherapy in patients with MPC.
After:
Pancreatic cancer is a lethal malignancy and the fourth leading cause of cancer-related death in the United States, with a current 5-year survival rate of only approximately 9%.[1] In up to 85% of patients, pancreatic cancer is diagnosed at an advanced stage because of infiltration of the surrounding vessels or distant metastasis.[2] In patients with metastatic pancreatic cancer (MPC), the combination of 5-fluorouracil (5-FU), leucovorin, irinotecan, and oxaliplatin (FOLFIRINOX) or gemcitabine and nab-paclitaxel resulted in significantly longer overall survival (OS) than that associated with gemcitabine monotherapy.[3,4] Furthermore, FOLFIRINOX has shown excellent efficacy in not only palliative but also adjuvant settings,[5] and this regimen has been used as the neoadjuvant chemotherapy in several phase III clinical trials.[6] The mechanism of FOLFIRINOX was regarded synergistic activity of irinotecan has when it is administered before fluorouracil and leucovorin and synergistic activity of irinotecan and oxaliplatin.[3] However, the regimen has been associated with a high incidence of adverse events including grade 3 or 4 neutropenia and fatigue.
The effects of gender in cancer treatment are not generally considered in preclinical experiments, clinical trials, or real clinical settings. However, there are differences in the efficacies and toxicities of chemotherapeutic agents between male and female patients.[7] For example, 5-FU, which is the backbone of the FOLFIRINOX regimen, degraded more slowly and was associated with higher toxicity in female patients.[8,9] In addition, a previous study reported that female gender may be a predictor of a better response to FOLFIRINOX. In this study, female gender was associated with a significantly higher disease control rate and showed a tendency towards a longer median progression-free survival (PFS).[10] However, a secondary analysis within the trial PRODIGE 4/ACCORD 11 did not conclusively show a possible effect of gender on the prognosis of patients receiving FOLFIRINOX because median OS and PFS in female was superior than those in male without statistical significance.[11] Therefore, the association remains controversial and elucidation is necessary for further evaluation of the effect of gender. The current study was aimed at investigating gender differences in clinical outcomes and dose modification patterns during FOLFIRINOX chemotherapy in patients with MPC.
